# Locally Differentially Private Document Generation Using Zero Shot Prompting

**Saiteja Utpala**
Cohere For AI
saitejautpala@gmail.com

**Sara Hooker**
Cohere For AI
sarahooker@cohere.com

**Pin Yu Chen**
IBM Research
pin-yu.chen@ibm.com

## Abstract

Numerous studies have highlighted the privacy risks associated with pretrained large language models. In contrast, our research offers a unique perspective by demonstrating that pretrained large language models can effectively contribute to privacy preservation. We propose a locally differentially private mechanism called DP-Prompt, which leverages the power of pretrained large language models and zero-shot prompting to counter author de-anonymization attacks while minimizing the impact on downstream utility. When DP-Prompt is used with a powerful language model like ChatGPT (gpt-3.5), we observe a notable reduction in the success rate of de-anonymization attacks, showing that it surpasses existing approaches by a considerable margin despite its simpler design. For instance, in the case of the IMDB dataset, DP-Prompt (with ChatGPT) perfectly recovers the clean sentiment F1 score while achieving a 46% reduction in author identification F1 score against static attackers and a 26% reduction against adaptive attackers. We conduct extensive experiments across six open-source large language models, ranging up to 7 billion parameters, to analyze various effects of the privacy-utility tradeoff. Code is avaliable at https://github.com/SaitejaUtpala/dp_prompt

## 1 Introduction

The vast amount of online text data has the potential to reveal numerous user attributes, making individuals easily identifiable (Rao et al., 2000; Hovy et al., 2015; Preoţiuc-Pietro et al., 2015). While private information can be directly disclosed through specific phrases in the text, it can also be implicitly inferred. For instance, linguistic patterns embedded within the text can inadvertently facilitate authorship attribution (Kešelj et al., 2003; Shrestha et al., 2017), leading to unintended privacy leakage.

An illustrative real-world scenario is the AOL search data leak in August 2006 (Pass et al.,

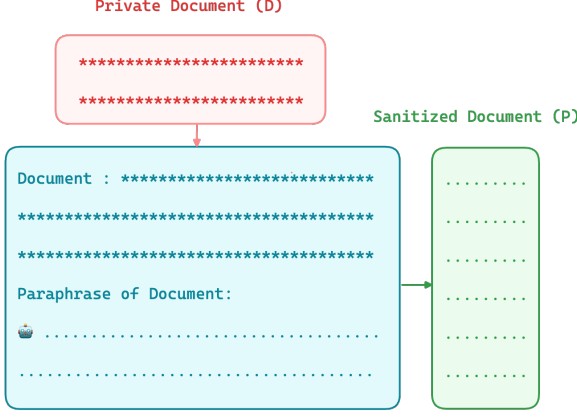

Figure 1: Overview of the proposed DP-Prompt mechanism. Given a private document (D), DP-Prompt generates a sanitized document (P) while ensuring local differential privacy. The level of privacy guarantee is controlled by adjusting the sampling temperature (T) during the decoding process.

2006). The incident unfolded when AOL mistakenly released detailed search logs of their users, wrongly assuming that the data had been adequately anonymized through the use of random user IDs. Unfortunately, the released logs contained sufficient personally identifiable information, leading to the identification of numerous individuals (Barbaro and Jr., 2006; Jones et al., 2007). This breach of privacy triggered widespread public outcry and led to the initiation of class action lawsuits.

This case is just one among many that highlights the limitations of ad-hoc privacy approaches that may give the impression of providing privacy but ultimately fall short. Differential privacy (DP) provides a rigorous treatment for the notion of data privacy by providing plausible deniability by precisely quantifying the deviation in the model's output distribution under modification of a small number of data points (Dwork et al., 2006, 2014). The provable guarantees offered by DP, coupled with its compelling properties such as immunity to arbi-

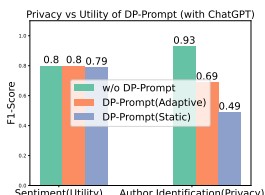
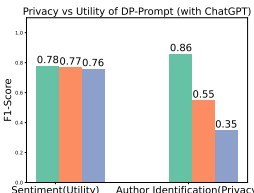

(a) IMDB (embedding access)   (b) Yelp (embedding access)

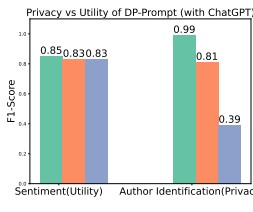
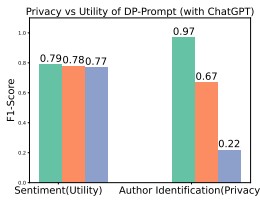

(c) IMDB (text access)   (d) Yelp (text acccess)

Figure 2: Overview of the privacy-utility tradeoff with DP-Prompt (using ChatGPT) for the IMDB and Yelp datasets, conducted at a temperature of 1.5. The terms 'Static' and 'Adaptive' refer to the attack models defined in Definition 2.

trary post-processing and graceful composability, have established it as the de facto standard for privacy. DP has witnessed widespread adoption and numerous deployments in both private (Erlingsson et al., 2014; Apple, 2017; Near, 2018) and public organizations (Abowd, 2018).

To address the issue of deanonymization attacks, various approaches have been proposed within the DP framework. These approaches encompass word-level strategies (Feyisetan et al., 2020; Xu et al., 2020; Carvalho et al., 2021) where noise is added at the word level, as well as sentence-level techniques (Meehan et al., 2022) where noise is added at the sentence level. However, recent research by (Mattern et al., 2022b) has identified limitations in word-level approaches, particularly their disregard for contextual information. To overcome these limitations, Mattern introduced a mechanism that fine-tunes the GPT-2 model (Radford et al., 2019) specifically for paraphrasing tasks, resulting in the generation of sanitized versions of documents. While promising, the approach is limited by their reliance on annotated paraphrasing data, extensive computing resources for larger models, and the quality of annotations.

We propose **DP-Prompt**, a novel and straightforward solution to address deanonymization attacks. Our method leverages pretrained large language models by directly prompting them to generate paraphrases. These paraphrases are then released

as sanitized documents in a zero-shot manner (See Figure 1). Our motivation for this approach stems from two important factors. Firstly, recent research (Bevendorff et al., 2019; Mattern et al., 2022b) has shown that paraphrasing is a robust defense mechanism against deanonymization attacks. Secondly, growing evidence suggests that pretrained large language models can effectively tackle complex tasks without the need for task-specific and expensive fine-tuning (Brown et al., 2020; Chowdhery et al., 2022; Chung et al., 2022; Kojima et al., 2022; OpenAI, 2023), through zero-shot prompting.

By harnessing the capabilities of pretrained large language models, DP-Prompt offers a straightforward and powerful solution to mitigate the risk of deanonymization. It provides a promising alternative that can be widely applicable, particularly in the context of on-device large language models where text completion tasks require significantly fewer resources. We summarize the contributions as follows:

- We propose DP-Prompt, a new, simple, and computationally effective differentially private (DP) mechanism designed as a defense against de-anonymization attacks. DP-Prompt takes a private document and generates a paraphrased version using zero-shot prompting. The resulting paraphrased document is then released as a sanitized document, as illustrated in Figure 1.

- We demonstrate that DP-Prompt, in combination with ChatGPT (gpt-3.5), surpasses all current methods in terms of utility for any level of privacy. Our approach successfully recovers clean sentiment F1 score while significantly reducing the accuracy of author deanonymization attacks. Refer to Figure 2 for an overview of these results.

- To demonstrate the broad applicability of DP-prompt, We conduct extensive experiments with 6 open source models ranging upto 7 billion parameters to study the privacy-utiliy tradeoff.

## 2 Preliminaries

A mechanism $\mathcal{M} : \mathcal{D} \to \mathcal{V}$ achieves $\epsilon$-PureDP if, for all inputs $D, D' \in \mathcal{D}$ that differ in one element, and for all $V \subseteq \text{Range}(\mathcal{M})$, $\Pr[\mathcal{M}(D) \in V] \le \exp(\epsilon)\Pr[\mathcal{M}(D') \in V]$ (Dwork et al., 2006).

| Mechanism | Privacy Level | Requires fine-tuning | Generates sanitized doc |
|---|---|---|---|
| Madlib (Feyisetan et al., 2020) | Word level Metric-DP | No | Yes |
| Mahanolbis (Xu et al., 2020) | Word level Metric-DP | No | Yes |
| TEM (Carvalho et al., 2021) | Word level Metric-DP | No | Yes |
| Truncated Laplace (Meehan et al., 2022) | Sentence level Pure-DP | No | No |
| Deep Candidate (Meehan et al., 2022) | Sentence level Pure-DP | Yes | No |
| Paraphraser (Mattern et al., 2022b) | Document level Pure-LDP | Yes | Yes |
| DP Prompt (Ours) | Document level Pure-LDP | No | Yes |

Table 1: We compare our proposed method, DP-Prompt, with related work on various factors. The "Privacy level" indicates the privacy guarantee provided by each mechanism. "Fine-tuning" denotes whether the mechanism involves fine-tuning a model as an intermediate step. The last column, "Generates sanitized doc," indicates whether the mechanism can output a fully sanitized document instead of just sanitized embeddings.

Metric Differential Privacy (Metric-DP) (Andrés et al., 2013; Chatzikokolakis et al., 2013) is a relaxation of Pure-DP that applies to data represented in a general metric space. For a given distance metric $d : \mathcal{D} \times \mathcal{D} \to \mathbb{R}_+$, a mechanism $\mathcal{M} : \mathcal{D} \to \mathcal{V}$ achieves $\epsilon d$-MetricDP if, for any $D, D' \in \mathcal{D}$ and for all $V \subseteq \text{Range}(\mathcal{M})$, $\Pr[\mathcal{M}(D) \in V] \leq \exp(d(D, D'))\Pr[\mathcal{M}(D') \in V]$.

Local differential privacy (LDP) (Kasiviswanathan et al., 2011; Duchi et al., 2013; Xiong et al., 2020) is a privacy framework where data is locally perturbed before transmission, considering the presence of an untrusted data collector or server. The formal definition of LDP is as follows:

**Definition 1** (PureLDP). *A randomized mechanism $\mathcal{M} : \mathcal{D} \to \mathcal{V}$ is said to be $\epsilon$-PureLDP if for any pair of inputs $D, D' \in \mathcal{D}$ and for all $V \subseteq \text{Range}(\mathcal{M})$*

$$\Pr[\mathcal{M}(D) \in V] \leq \exp(\epsilon)\Pr[\mathcal{M}(D') \in V].$$

There is a growing consensus that, despite the assurance of formal guarantees, it is imperative to subject differentially private mechanisms to robust privacy attacks that simulate strong and malicious adversaries (Jayaraman and Evans, 2019; Blanco-Justicia et al., 2022). Such evaluation allows to effectively assess the empirical privacy provided by the mechanism in real-world scenario. To this end we define four attack models depending its adaptivity and mode of access.

**Definition 2** (Attack Models). *Consider a collection of private documents $(D_1, \ldots, D_n)$ from distribution $\mathcal{D}$ with associated author identities $(a_1, \ldots, a_n)$ and embeddings $(E_1, \ldots, E_n) \sim \mathcal{E}$.*

*For text-to-text sanitization using mechanism $\mathcal{M}_{\text{text}}$, the sanitized documents are represented as $(P_1, \ldots, P_n) \sim \mathcal{P}_{\mathcal{M}_{\text{text}}}$. For text-to-embedding sanitization via mechanism $\mathcal{M}_{\text{embedding}}$, the sanitized embeddings are denoted as $(N_1, \ldots, N_n) \sim \mathcal{N}_{\mathcal{M}_{\text{embedding}}}$*

- **Static Attacker with Embedding Access**: *Has access to clean documents $(D_1, \ldots, D_n)$ but lacks access to sanitized versions $(P_1, \ldots, P_n)$.*

- **Static Attacker with Text Access** : *Doesn't have access to sanitized embeddings $(N_1, \ldots, N_n)$ but only to the clean embeddings $(E_1, \ldots, E_n)$.*

- **Adaptive Attacker with Embedding Access** : *Has access to sanitized embeddings $(N_1, \ldots, N_n)$. Hence, trains a de-anonymization model to adapt to the DP mechanism $\mathcal{M}_{\text{embedding}}$.*

- **Adaptive Attacker with Text Access**: *Has access to sanitized text $(P_1, \ldots, P_n)$. Consequently, trains a de-anonymization model to adapt to the DP mechanism $\mathcal{M}_{\text{text}}$.*

It is important to note that the adaptive attacker is a more formidable adversary since it adapts to the characteristics of the mechanism $\mathcal{M}$, whereas the static attacker only has access to clean documents/clean embeddings without any added noise. The mode of access—either raw text or abstracted embeddings—offers further nuances, determining the exact nature of the data an attacker can exploit.

## 3 DP Prompt

Language models use a decoder network to generate sequential text output. Given a context or prompt represented by a sequence of tokens $C = (c_1, \ldots, c_m)$, the language model generates text by sampling tokens from a conditional distribution $\Pr_{|C}(x_1, \ldots, x_n) = \prod_{i=1}^{n} \Pr_{|C}(x_i | x_1, \ldots, x_{i-1})$. In this distribution, the logits $\mathbf{u} \in \mathbb{R}^{|\mathcal{V}|}$ are transformed using the softmax function with a temperature $T$, where $p_{ij} = \frac{\exp(\frac{u_{ij}}{T})}{\sum_{j=1}^{|\mathcal{V}|} \exp(\frac{u_{ij}}{T})}$, and $\mathcal{V}$ represents the vocabulary.

Figure 3: Sample illustration of clean and sanitized documents for various mechanisms. For ChatGPT, prompt is "Review:[review]Paraphrase of the Review:" where review is the clean review.

This process of sequentially generating text can be regarded as a problem of selecting tokens at each step. Hence, to make the generation step differentially private, one must replace it with a differentially private version of the selection process. One commonly used and well-known differentially private mechanism is the exponential mechanism (McSherry and Talwar, 2007), which is defined as follows:

**Definition 3** (Exponential Mechanism). *Given an utility function* $u : \mathcal{D} \times \mathcal{V} \to \mathcal{V}$. *The exponential mechanism* $\mathcal{M}_{\text{Exp}} : \mathcal{D} \to \mathcal{V}$ *is a randomized algorithm with output distribution* $P\left[\mathcal{M}_{\text{Exp}}(D) = v\right] \propto \exp\left(\frac{\epsilon u(D,v)}{2\Delta u}\right)$, *where* $\Delta u = \max_{D,D',v} |u(D,v) - u(D',v)|$ *is sensitivity.*

In our case, the utility of token $v_j \in \mathcal{V}$ at each step $i$ is simply the logit $u_{ij} \in \mathbb{R}$. Hence, one can make text generation differentially private using the exponential mechanism.

Extensive research has shown that paraphrasing documents helps conceal author identity (Rao et al., 2000; Bevendorff et al., 2019; Mattern et al., 2022b). Considering recent advancements where tasks are formulated as prompts and language models are tasked to complete them (Raffel et al., 2020; Brown et al., 2020; Wei et al., 2022), we directly prompt the language model to generate paraphrases. Therefore, given a private document D and a specific prompt template instructing the language model to generate a paraphrase, such as T := "Paraphrase of the document:" we combine D and T to create a context C. By utilizing this context, we execute the text generation procedure in a differentially private manner to produce a paraphrase. We refer to this procedure as DP-Prompt. Algorithm 1 outlines the specific steps of our pro-

---

**Algorithm 1: DP-Prompt**

**Input:** language model (LM), private document (D), prompt template (T), clipping vector $\mathbf{b} \in \mathbb{R}^{|\mathcal{V}|}$, temperature $T \in \mathbb{R}_+$, paraphrase tokens n.
**Output:** Sanitized Doc (P)

1 $P \leftarrow []$, $C \leftarrow \text{GeneratePrompt}(D, T)$
2 $C_{\text{tokens}} \leftarrow \text{Tokenize}(C)$
3 **for** $i \leftarrow 1$ **to** $n$ **do**
4      $\mathbf{u} \leftarrow \text{LM}(C_{\text{tokens}})$
5      $\mathbf{u}' \leftarrow \text{ClipAndScale}(\mathbf{u}, \mathbf{b}, T)$
6      $\mathbf{p} \leftarrow \text{ConvertToProbabilities}(\mathbf{u}')$
7      $v \leftarrow \text{SampleToken}(\mathbf{p})$
8      $P \leftarrow P \cup [v]$, $C_{\text{tokens}} \leftarrow C_{\text{tokens}} \cup [v]$
9 **end**
10 $P \leftarrow \text{Detokenize}(P)$ **return** P

---

posed DP-Prompt, and the code for implementing DP-Prompt in HuggingFace (Wolf et al., 2019) is provided in Appendix B. The formal guarantee of achieving $\epsilon$-PureLDP is provided by the following theorem:

**Theorem 1.** *Suppose the language model has not been pretrained on the private documents distribution* $\mathcal{D}$. *If the final logits* $\mathbf{u} \in \mathbb{R}^{|\mathcal{V}|}$ *satisfy the condition* $b_1 \leq u_i \leq b_2, \forall i$, *and the* DP-Prompt *run with a temperature* $T$ *for generating* $n$ *tokens, then it can be proven that the generated output satisfies* $(2n(b_2 - b_1)/T)$-LDP.

See the Appendix A for the proof.

## 4 Experiments

### 4.1 Experiment Setup

**Evaluation:** Note that we are comparing DP-mechanisms with different levels of differential privacy. Therefore, in our experiments, we focus on evaluating the empirical privacy rather than the theoretical privacy($\epsilon$) for effective and realistic as-

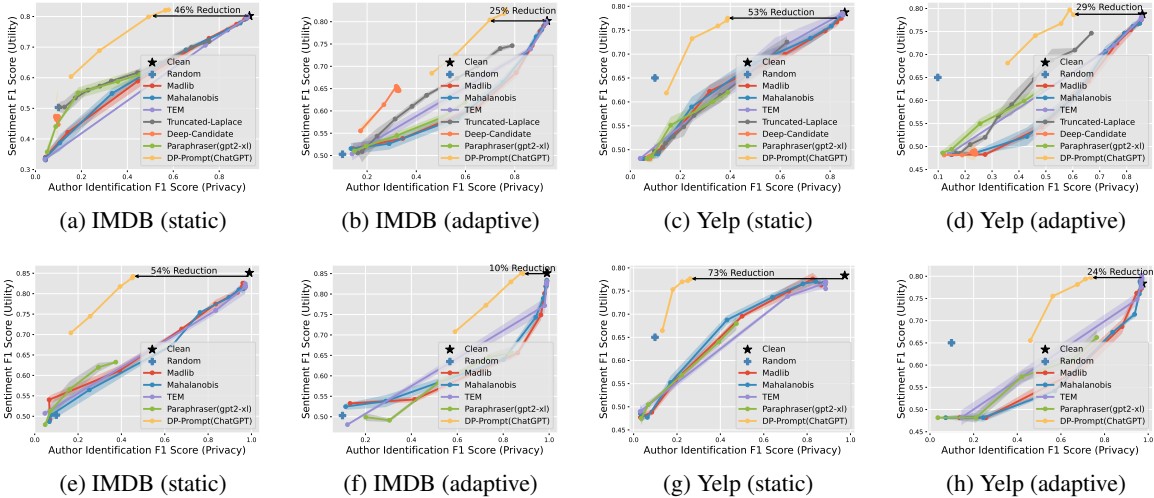

| (a) IMDB (static) | (b) IMDB (adaptive) | (c) Yelp (static) | (d) Yelp (adaptive) |

| (e) IMDB (static) | (f) IMDB (adaptive) | (g) Yelp (static) | (h) Yelp (adaptive) |

Figure 4: Comparison of DP-Prompt (with ChatGPT) with various baselines. The top row shows results for an attacker with embedding access, while the row below presents results for an attacker with text access. Notably, it is evident that regardless of the chosen privacy level, DP-Prompt, when utilized with ChatGPT (GPT-3.5), exhibits significantly better utility compared to all baseline mechanisms.

sessment. As a result, we plot the author identification F1 score, which is calculated by conducting de-anonymization attacks on the sanitized documents. This score indicates the potential for privacy breaches. On the other hand, the y-axis represents the sentiment F1 score, which measures the utility of the sanitized documents.

**Datasets:** We conduct experiments using IMDB movie reviews and Yelp business reviews, both of which contain author and sentiment labels. The IMDB dataset has a size of 15,000, while the Yelp dataset has 17,336 samples. For both datasets, sentiment analysis is a 2-class classification task, and the author identification task is a 10-class classification task.

**Implementation Details:** For the embedding-level attacker, we utilize 3-Layer MLPs with ReLU activation functions and train them on sentence embeddings (Reimers and Gurevych, 2019). For the text-level attacker, we fine-tune BERT (Devlin et al., 2018). More details can be found in Appendix C. Regarding the static attacker, the clean set of documents is used for training and validation, while the sanitized documents serve as the test set. On the other hand, for the adaptive attacker, all three sets (training, validation, and testing) consist of sanitized documents.

- For each of word level mechanisms, (Madlib (Feyisetan et al., 2020), Mahalanobis (Xu et al., 2020), TEM (Carvalho et al., 2021)) we run the mechanisms for 8 $\epsilon$'s given $\epsilon =$

$$\{2, 5, 8, 11, 14, 17, 20, 25\}$$

- For each of sentence level mechanisms (Truncated-Laplace (Meehan et al., 2022), Deep-Candidate (Meehan et al., 2022)): we run the mechanisms for 11 $\epsilon$'s given by $\epsilon = \{5, 10, 20, 30, 40, 50, 75, 100, 150, 200\}$.

- For Paraphraser (Mattern et al., 2022b) and DP-Prompt with open source models we run decoding at 5 temperatures $\{0.75, 1.0, 1.25, 1.5, 1.75\}$. For DP-Prompt we run ChatGPT at temperatures $\{1.0, 1.25, 1.5, 1.75, 2.0\}$.

Further we also consider F1 scores on Clean (without noise added) embeddings/documents and performance of uniformly random classifier (for more details, refer to Appendix C.6).

## 4.2 DP-prompt with ChatGPT (gpt-3.5)

In this section we compare 6 baselines (Madlib, Mahalanobis, Tem, Truncated-laplace, Deep-candidate, Paraphraser) run with configurations above (for more details also refer to Appendix C) with DP-Prompt with ChatGPT. Except for DP-Prompt, we run each mechanism to 3 times to produce 3 different sanitized documents and plot mean author F1 identification score on x-axis and show $2\sigma$ band around mean sentiment F1 score. Results are show in Figure 4

The results clearly demonstrate the superior performance of DP-Prompt with ChatGPT (GPT-3.5).

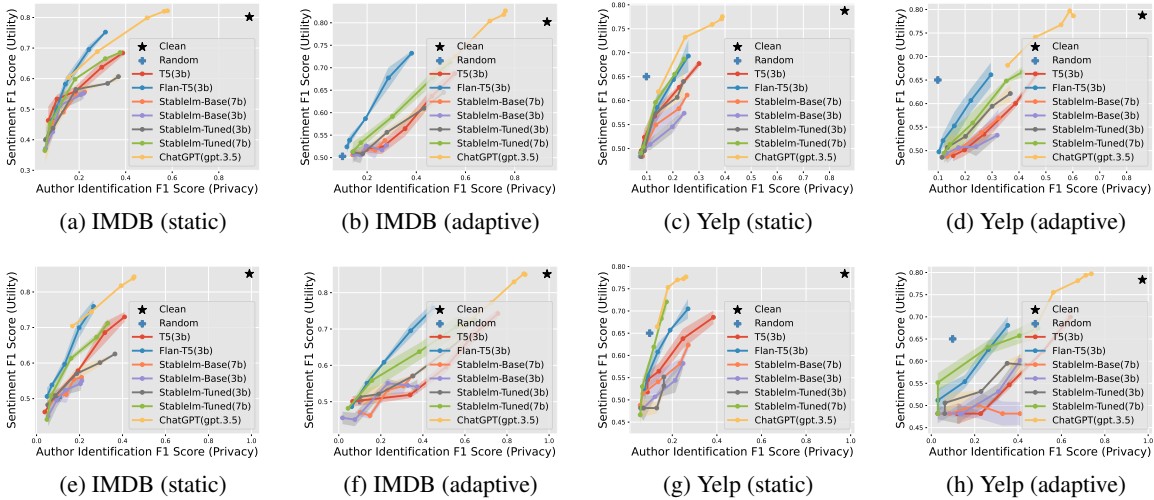

| | | | |
|---|---|---|---|
| (a) IMDB (static) | (b) IMDB (adaptive) | (c) Yelp (static) | (d) Yelp (adaptive) |
| (e) IMDB (static) | (f) IMDB (adaptive) | (g) Yelp (static) | (h) Yelp (adaptive) |

Figure 5: Illustration of privacy-utility tradeoff in DP-Prompt with open source models and ChatGPT(gpt-3.5). The top row shows results for an attacker with embedding access, while the row below presents results for an attacker with text access.

Notably, DP-Prompt exhibits significantly higher utility on the y-axis for a chosen empirical privacy value on the x-axis. All word-level mechanisms show a similar privacy-utility tradeoff. Regarding sentence-level mechanisms, the truncated Laplace mechanism performs decently, while in the static attack experiments, Deep-candidate is reduced to a random classifier due to the distribution shift caused by sentence recoding.

Furthermore, in the case of clean reviews (i.e., without any noise), the embedding-level attacker can accurately identify the author among 10 different options with a high F1 score of 0.93 in IMDB and 0.86 in Yelp. However, when DP-Prompt is employed, the sentiment F1 scores remain unchanged, while the author identification scores decrease by 46% and 25% in the case of IMDB, and 53% and 29% in the case of Yelp.

The text-level models are more accurate than the embedding-level models, with author identification scores of 0.99 (as opposed to 0.93) and 0.97 (as opposed to 0.86) in IMDB and Yelp, respectively, for clean reviews. When DP-Prompt is employed, the sentiment F1 scores remain unchanged, while the author identification scores decrease by 54% and 10% in the case of IMDB, and 73% and 24% in the case of Yelp. This illustrates that text-level attackers are more powerful.

## 4.3 DP-Prompt with open source models

It is important to note that models such as ChatGPT (gpt-3.5) are proprietary and can only be accessed through APIs, necessitating the uploading of user documents to the language model provider. Although DP-Prompt with such proprietary models provide LDP guarantee, this defeats the fundamental motivation of LDP, which is to achieve privacy guarantees without relying on a trusted data curator. The objective of the experiments presented in the preceding section aim to demonstrate that DP-Prompt, when combined with a powerful language model like ChatGPT, can outperform existing methods by a significant margin.

Considering the increasing interest in building high-quality open-source large language models (Scao et al., 2022; Black et al., 2022; Touvron et al., 2023; Li et al., 2023; Jiang et al., 2023), we expand our evaluation of DP-Prompt to include six open-source models, ranging in size up to seven billion parameters. Our evaluation takes into account two factors: (i) architecture, where we consider both encoder-decoder and decoder-only models, and (ii) the level of fine-tuning, including models that are fine-tuned using instructions and/or Reinforcement Learning with Human Feedback (RLHF). Specifically 6 models are as follows, *Base*: T5 (3b), Stable lm base (3b, 7b),*Instruction finetuned/RLHF tuned:* Flan T5 (3b), Stable lm tuned (3b,7b).

While T5, Flan T5 are encoder-decoder models, rest of the models are decoder-only. In contrast to DP-Prompt with ChatGPT, we perform DP-Prompt using the aforementioned open source language models three times for each dataset and temperature, resulting in three sanitized documents. Fur-

| | | Data | IMDB | | | | | | | | | | Yelp | | | | | | | | | |
|---|---|---|---|---|---|---|---|---|---|---|---|---|---|---|---|---|---|---|---|---|---|---|
| | | Metric | Sentiment F1 score | | | | | Author Identification F1 Score | | | | | Sentiment F1 score | | | | | Author Identification F1 Score | | | | |
| | | clipping | 0.75 | 1.0 | 1.25 | 1.5 | 1.75 | 0.75 | 1.0 | 1.25 | 1.5 | 1.75 | 0.75 | 1.0 | 1.25 | 1.5 | 1.75 | 0.75 | 1.0 | 1.25 | 1.5 | 1.75 |
| Flan-t5 (3b) | Static Attacker | Yes | 0.74 | 0.67 | 0.56 | 0.45 | 0.38 | 0.26 | 0.21 | 0.13 | 0.07 | 0.05 | 0.69 | 0.62 | 0.58 | 0.54 | 0.48 | 0.21 | 0.17 | 0.12 | 0.09 | 0.07 |
| | | No | 0.75 (+0.01) | 0.69 (+0.02) | 0.58 (+0.02) | 0.46 (+0.01) | 0.40 (+0.02) | 0.31 (+0.05) | 0.24 (+0.03) | 0.14 (+0.01) | 0.08 (+0.01) | 0.05 (+0.00) | 0.69 (+0.00) | 0.64 (+0.02) | 0.58 (+0.00) | 0.54 (+0.00) | 0.49 (+0.01) | 0.25 (+0.04) | 0.20 (+0.03) | 0.13 (+0.01) | 0.09 (+0.00) | 0.07 (+0.00) |
| | Adaptive Attacker | Yes | 0.72 | 0.66 | 0.58 | 0.52 | 0.51 | 0.34 | 0.25 | 0.16 | 0.13 | 0.11 | 0.67 | 0.59 | 0.54 | 0.52 | 0.49 | 0.25 | 0.20 | 0.14 | 0.11 | 0.10 |
| | | No | 0.73 (+0.01) | 0.67 (+0.01) | 0.58 (+0.00) | 0.53 (+0.01) | 0.52 (+0.01) | 0.38 (+0.04) | 0.28 (+0.03) | 0.19 (+0.03) | 0.13 (+0.00) | 0.11 (+0.00) | 0.67 (0.00) | 0.60 (+0.01) | 0.55 (+0.01) | 0.52 (+0.00) | 0.49 (+0.00) | 0.29 (+0.04) | 0.22 (+0.02) | 0.15 (+0.01) | 0.11 (+0.01) | 0.10 |
| Stablelm Tuned (7b) | Static Attacker | Yes | 0.67 | 0.63 | 0.53 | 0.38 | 0.34 | 0.33 | 0.29 | 0.12 | 0.05 | 0.03 | 0.66 | 0.62 | 0.55 | 0.49 | 0.48 | 0.28 | 0.22 | 0.10 | 0.08 | 0.07 |
| | | No | 0.68 (+0.01) | 0.66 (+0.03) | 0.59 (+0.06) | 0.45 (+0.07) | 0.36 (+0.02) | 0.37 (+0.04) | 0.31 (+0.02) | 0.18 (+0.06) | 0.07 (+0.02) | 0.05 (+0.02) | 0.68 (+0.02) | 0.65 (+0.03) | 0.59 (+0.04) | 0.50 (+0.01) | 0.49 (+0.01) | 0.26 (-0.02) | 0.21 (-0.01) | 0.13 (+0.03) | 0.08 (+0.00) | 0.07 (+0.00) |
| | Adaptive Attacker | Yes | 0.65 | 0.60 | 0.54 | 0.51 | 0.50 | 0.45 | 0.37 | 0.22 | 0.14 | 0.12 | 0.63 | 0.57 | 0.50 | 0.49 | 0.48 | 0.36 | 0.28 | 0.17 | 0.13 | 0.10 |
| | | No | 0.71 (+0.06) | 0.67 (+0.07) | 0.59 (+0.05) | 0.53 (+0.02) | 0.51 (0.01) | 0.53 (+0.08) | 0.46 (+0.09) | 0.30 (+0.02) | 0.17 (+0.03) | 0.14 (+0.02) | 0.66 (+0.03) | 0.64 (+0.07) | 0.55 (+0.05) | 0.51 (+0.02) | 0.49 (0.01) | 0.41 (+0.05) | 0.35 (+0.07) | 0.22 (+0.05) | 0.14 (+0.01) | 0.12 (+0.02) |

Table 2: Effect of clipping (shown as Yes) and without clipping (shown as No) on privacy-utility tradeoff.

ther we all run these open source models at half precision (torch.float16) and with max number of tokens in paraphrase to 150. The obtained results are presented in Figure 5.

Now we compare open source models along various factors

**Base vs Instruction/RLHF tuned:** It is evident that the base models perform poorly in comparison to the models that underwent instruction fine-tuning/RLHF tuning. One important point to mention is that both StableLM-Based (3b, 7b) perform significantly worse against adaptive attackers.

**Scale:** We observe that scale plays a key role, as StableLM-tuned (7b) demonstrates better utility than StableLM-Tuned (3b) across all levels of empirical privacy, with ChatGPT outperforming both.

**Variance:** It is worth noting that we did not obtain variance for DP-Prompt with ChatGPT in Figure 4 through multiple runs. However, our analysis in Figure 5 suggests that there is no significant variance in the Sentiment F1 score, at least for open source models.

**Encoder-Decoder / Decoder only:** Flan-T5(3b), an encoder-decoder model, outperforms the larger Stablelm-Tuned (7b) model. Notably, Flan-T5(3b) is fine-tuned exclusively on academic datasets, resulting in shorter paraphrases compared to Stablelm-Tuned (7b).

**ChatGPT vs Rest:** While the open-source models we considered demonstrate competitiveness with ChatGPT, a notable gap remains. The key finding is that even at a higher temperature of 1.5, Chat-GPT is capable of recovering a clean sentiment F1 score, while none of the open-source models can achieve a matching clean sentiment F1 score, even at a significantly lower temperature of 0.75. See Figure 7, which presents the paraphrases output by ChatGPT and Stablelm-Tuned (7B).

### 4.4 Effect of clipping logits

In both sections (Section 4.2 and Section 4.3) DP-Prompt is run without logit clipping. This is primar-

ily because ChatGPT doesn't expose logits, and for a fair comparison with ChatGPT, we didn't clip logits for open-source models in Section 4.3. However, the LDP guarantee still holds because, in practice, logits are always bounded within a certain precision, even though they may have large values.

In this section, we examine the impact of logit clipping on the tradeoff between privacy and utility using FlanT5(3b) and Stablelm Tuned(7b) models against embedding-level attacks. We adopt the approach of (Li and Clifton, 2021) by learning the clipping boundaries on additional data (more details can be found in Appendix C). The results of this analysis are presented in Table 2. The maximum difference observed with and without clipping occurs for Stablelm-Tuned(7b) on the IMDB dataset at a temperature of 0.75. In this case, the sentiment F1 score drops from 0.71 to 0.65, and the author identification F1 score drops from 0.53 to 0.45. Based on these findings, we recommend using logit clipping when higher privacy is required and not using clipping when higher utility is desired.

### 4.5 Effect of Top-K sampling

For the differential privacy guarantee to hold, sampling from probabilities must be done over the entire vocabulary according to score probabilities. While it is common to use top-k sampling in practice to improve generation (the default value in Hugging Face (Wolf et al., 2019) is 40). It is important to note that the ChatGPT chat completion API does not include a top-k parameter. In this section, using open-source models, we examine the impact of top-k sampling on utility and investigate whether it provides any empirical privacy, even in the *absence* of the differential privacy guarantee. In addition, we aim to assess whether top-k sampling can effectively help open source models to recover the clean sentiment F1-score and narrow the gap compared to DP-Prompt when used with ChatGPT.

We consider Flan-T5(3b) and Stablelm-

| | | Data Metric top_k | IMDB Sentiment F1 score | | | | | IMDB Author Identification F1 Score | | | | | Yelp Sentiment F1 score | | | | | Yelp Author Identification F1 Score | | | | |
|---|---|---|---|---|---|---|---|---|---|---|---|---|---|---|---|---|---|---|---|---|---|---|
| | | | 0.75 | 1.0 | 1.25 | 1.5 | 1.75 | 0.75 | 1.0 | 1.25 | 1.5 | 1.75 | 0.75 | 1.0 | 1.25 | 1.5 | 1.75 | 0.75 | 1.0 | 1.25 | 1.5 | 1.75 |
| Flan-t5 | Static Attacker | all | 0.75 | 0.69 | 0.58 | 0.46 | 0.40 | 0.31 | 0.24 | 0.14 | 0.08 | 0.05 | 0.69 | 0.64 | 0.58 | 0.51 | 0.49 | 0.25 | 0.20 | 0.13 | 0.09 | 0.07 |
| | | 80 | 0.75 (+0.0) | 0.72 (+0.03) | 0.70 (+0.12) | 0.69 (+0.23) | 0.66 (+0.26) | 0.31 (+0.0) | 0.27 (+0.03) | 0.23 (+0.09) | 0.22 (+0.14) | 0.21 (+0.16) | 0.71 (+0.02) | 0.69 (+0.05) | 0.68 (+0.10) | 0.64 (+0.13) | 0.64 (+0.15) | 0.25 (+0.00) | 0.20 (+0.00) | 0.19 (+0.06) | 0.19 (+0.10) | 0.17 (+0.10) |
| | | 40 | 0.75 (+0.00) | 0.74 (+0.24) | 0.72 (+0.23) | 0.69 (+0.27) | 0.67 (+0.27) | 0.32 (+0.0) | 0.29 (+0.05) | 0.26 (+0.12) | 0.25 (+0.17) | 0.23 (+0.18) | 0.71 (+0.02) | 0.69 (+0.05) | 0.68 (+0.10) | 0.64 (+0.13) | 0.64 (+0.15) | 0.25 (+0.00) | 0.22 (+0.02) | 0.20 (+0.07) | 0.19 (+0.10) | 0.17 (+0.10) |
| | Adaptive Attacker | all | 0.73 | 0.67 | 0.58 | 0.53 | 0.52 | 0.38 | 0.28 | 0.19 | 0.12 | 0.11 | 0.66 | 0.60 | 0.55 | 0.52 | 0.49 | 0.29 | 0.22 | 0.15 | 0.11 | 0.10 |
| | | 80 | 0.76 (+0.03) | 0.70 (+0.03) | 0.69 (+0.11) | 0.66 (+0.13) | 0.65 (+0.13) | 0.39 (+0.01) | 0.33 (+0.05) | 0.29 (+0.10) | 0.27 (+0.15) | 0.26 (+0.15) | 0.67 (+0.01) | 0.66 (+0.06) | 0.62 (+0.07) | 0.60 (+0.08) | 0.60 (+0.11) | 0.31 (+0.02) | 0.24 (+0.02) | 0.23 (+0.08) | 0.20 (+0.09) | 0.20 (+0.10) |
| | | 40 | 0.75 (+0.02) | 0.72 (+0.05) | 0.70 (+0.12) | 0.67 (+0.14) | 0.67 (+0.15) | 0.41 (+0.02) | 0.36 (+0.08) | 0.33 (+0.14) | 0.31 (+0.19) | 0.29 (+0.19) | 0.69 (+0.02) | 0.66 (+0.06) | 0.65 (+0.10) | 0.63 (+0.11) | 0.61 (+0.12) | 0.31 (+0.02) | 0.26 (+0.04) | 0.24 (+0.09) | 0.23 (+0.12) | 0.23 (+0.13) |
| Stablelm Tuned (7b) | Static Attacker | all | 0.68 | 0.66 | 0.59 | 0.45 | 0.36 | 0.37 | 0.31 | 0.18 | 0.07 | 0.05 | 0.68 | 0.65 | 0.59 | 0.50 | 0.49 | 0.24 | 0.20 | 0.13 | 0.08 | 0.07 |
| | | 80 | 0.69 (+0.01) | 0.67 (+0.01) | 0.66 (+0.07) | 0.62 (+0.17) | 0.62 (+0.26) | 0.37 (+0.00) | 0.34 (+0.03) | 0.28 (+0.10) | 0.24 (+0.20) | 0.24 (+0.19) | 0.69 (+0.01) | 0.66 (+0.01) | 0.65 (+0.06) | 0.62 (+0.12) | 0.59 (+0.10) | 0.24 (+0.00) | 0.22 (+0.02) | 0.19 (+0.06) | 0.16 (+0.08) | 0.16 (+0.09) |
| | | 40 | 0.69 (+0.01) | 0.67 (+0.01) | 0.66 (+0.07) | 0.64 (+0.19) | 0.64 (+0.28) | 0.38 (+0.01) | 0.34 (+0.03) | 0.30 (+0.12) | 0.27 (+0.20) | 0.27 (+0.22) | 0.69 (+0.01) | 0.69 (+0.04) | 0.61 (+0.02) | 0.65 (+0.15) | 0.65 (+0.16) | 0.25 (+0.01) | 0.22 (+0.02) | 0.20 (+0.07) | 0.19 (+0.11) | 0.18 (+0.11) |
| | Adaptive Attacker | all | 0.70 | 0.67 | 0.59 | 0.53 | 0.51 | 0.53 | 0.46 | 0.30 | 0.17 | 0.14 | 0.66 | 0.64 | 0.55 | 0.51 | 0.49 | 0.41 | 0.35 | 0.22 | 0.14 | 0.12 |
| | | 80 | 0.70 (+0.00) | 0.68 (+0.01) | 0.67 (+0.08) | 0.64 (+0.11) | 0.63 (+0.12) | 0.53 (+0.00) | 0.49 (+0.03) | 0.44 (+0.14) | 0.40 (+0.23) | 0.35 (+0.21) | 0.69 (+0.03) | 0.66 (+0.02) | 0.64 (+0.09) | 0.59 (+0.08) | 0.60 (+0.11) | 0.41 (+0.00) | 0.38 (+0.03) | 0.32 (+0.10) | 0.29 (+0.15) | 0.27 (+0.15) |
| | | 40 | 0.70 (+0.00) | 0.69 (+0.02) | 0.67 (+0.08) | 0.64 (+0.11) | 0.63 (+0.12) | 0.53 (+0.00) | 0.49 (+0.03) | 0.45 (+0.15) | 0.42 (+0.25) | 0.39 (+0.25) | 0.69 (+0.03) | 0.67 (+0.03) | 0.64 (+0.09) | 0.60 (+0.09) | 0.62 (+0.13) | 0.42 (+0.01) | 0.37 (+0.02) | 0.34 (+0.12) | 0.32 (+0.18) | 0.29 (+0.17) |

Table 3: The effect of top-k sampling on privacy-utility tradeoff on Flan-T5(3b) and Stablelm-Tuned(7B) models

Tuned(7B) models and perform decoding with top-k sampling for $k = \{40, 80\}$. The results against embedding level attacks, including no top-k sampling, are shown in Table 3. The maximum difference observed with and without clipping occurs for Stablelm-Tuned(0.75) on the IMDB dataset at a temperature of 0.7 for $k = 40$. In this case, the sentiment F1 score increases from 0.32 to 0.64, and the author identification F1 score increases from 0.05 to 0.27. Additionally, top-k sampling does not improve utility at a temperature of 0.75, indicating that DP-Prompt with open-source models, even with top-k sampling, does not match DP-Prompt with ChatGPT.

## 5 Issue of Data Memorization

It is important to acknowledge the possibility that ChatGPT and other open-source language models (LLMs) may have been exposed to online review datasets such as IMDB and Yelp. This exposure is not unique to our approach. Similar questions can be raised for word-level approaches (Feyisetan et al., 2020; Xu et al., 2020; Carvalho et al., 2021) and sentence-level approaches (Meehan et al., 2022) since GloVe embeddings (Pennington et al., 2014) and Sentence Transformers (BERT) (Reimers and Gurevych, 2019) are pretrained on the Common Crawl dataset. Hence assumption in Theorem 1 may not hold. We argue that this exposure does not pose a major concern for DP-Prompt due to the following reasons.

**No availability of paraphrases of IMDB, Yelp:** DP-Prompt essentially evaluates the performance of LLMs in the task of zero-shot paraphrasing at higher temperatures. Although the language models may have been fine-tuned using open annotated paraphrase datasets (Zhang et al., 2019), it's important to note that there is no specific annotated

paraphrasing data available for the IMDB and Yelp datasets. Therefore, the language models used in DP-Prompt are not explicitly fine-tuned for paraphrasing tasks related to these datasets. Consequently, the results obtained in Section 4 are likely to generalize to a large extent.

**Robustness of Evaluation Setup:** Our evaluation methodology goes beyond relying solely on the Sentiment F1 score (Utility) or even comparing sentiment F1 score (Utility) with $\epsilon$ (theoretical privacy) as presented in (Meehan et al., 2022). Instead, we adopt a comprehensive approach that considers the merit of our proposed approach based on the sentiment F1 score in conjunction with empirical privacy demonstration against de-anonymization attacks. Simply copying reviews verbatim would yield a high author identification F1 score, failing to provide empirical privacy. Additionally, we consider not only static attack models but also stronger ones like adaptive attackers. Superficially altering reviews may deceive static attackers but would likely fail against adaptive attackers.

## 6 Related Work

The previous work on releasing private documents can be categorized into three approaches based on the level at which noise is added (See Table 1 for concise summary). These approaches are:

**Word-level Approaches**: MadLib (Feyisetan et al., 2020) is a word-level mechanism that applies Laplace noise to word embeddings and maps them to the nearest word in the vocabulary, demonstrating the differential privacy (DP) guarantee of MadLib under the Euclidean metric. An extension of this approach involves using a regularized Mahalanobis metric instead (Xu et al., 2020). In contrast, the TEM mechanism utilizes the exponential mechanism to transform the privatization step into a

selection problem (Carvalho et al., 2021). Furthermore, there is a recent development known as CusText (Chen et al., 2023), which focuses on developing customized mapping mechanisms for each individual word in the vocabulary (Chen et al., 2023). All of these approaches are word-level mechanisms and have been shown to have significant limitations, such as their disregard for contextual information (Mattern et al., 2022b).

**Sentence-level Approaches**: Sentence-level mechanisms based on Sentence Transformer (Reimers and Gurevych, 2019) were introduced in (Meehan et al., 2022). They proposed two approaches: one approach where noise is added to sentence embeddings, and another more complicated approach based on maximizing Tukey depth (Tukey, 1975; Gilad-Bachrach and Burges, 2012).

**Document-level Approaches**: A document-level Local Differential Privacy (LDP) mechanism was introduced, where GPT-2 is fine-tuned for a paraphrasing task (Mattern et al., 2022b). Our approach, DP-Prompt, draws inspiration from their work, but instead of resource-intensive fine-tuning, we use a zero-shot approach with pretrained models for efficient and effective generation of sanitized documents. Furthermore, the recently proposed DP-BART (Igamberdiev and Habernal, 2023) employs BART (Lewis et al., 2020), an encoder-decoder model. In DP-BART, noise is added to the encoder's output, and the decoder is fine-tuned to adapt to this noisy encoder output.

**Adversarial Methods:** Parallel to differentially private approaches, other techniques have been proposed that utilize Adversarial Learning (Shetty et al., 2018; Quiring et al., 2019) and Data Poisoning (Wang et al., 2022; Jin et al., 2020). However, these methods typically require access to a surrogate classifier. In contrast, our method is zero-shot, requiring neither fine-tuning nor access to a classifier.

**Differentially Private Training/Fine Tuning:** There is extensive research on differentially private training or fine-tuning of language models (Kerrigan et al., 2020; Li et al., 2021; Yu et al., 2021; Anil et al., 2022; Mattern et al., 2022a). They aim to make language models resistant to various kinds of data leakage attacks (Carlini et al., 2019, 2021; Deng et al., 2021; Balunovic et al., 2022). It is important to emphasize that this line of work is completely distinct from our own, as it focuses on training language models on private data, while our goal is to generate sanitized documents from private documents using pretrained language models.

## 7 Conclusion

This paper introduces DP-Prompt, a locally differentially mechanism called DP-Prompt that generates sanitized versions of private documents by prompting large language models to generate paraphrases. Notably, our method offers a simpler approach compared to existing methods. Through extensive experiments, we show that our approach achieves significantly improved utility compared to current methods for any required level of privacy. As the demand for on-device large language models (LLMs) continues to grow, our method emerges as a reliable safeguard for users' privacy and provides robust defense against de-anonymization attacks.

## 8 Limitations

In our study, we explored the initial step of harnessing large language models and zero-shot prompting to generate sanitized documents. While this approach effectively conceals the specific writing style of authors, there is still a potential risk of revealing explicit personal information, such as zip codes, bank details, or gender, especially when naively prompting a language model. This risk is particularly relevant in alternative text formats like messages or emails compared to online reviews.

For future work, an important direction would be to define a set of sensitive attributes and directly prompt the language model to replace these attributes with the identifier "X" while generating paraphrases. This approach would help improve the safeguarding of personal information. Additionally, it would be worthwhile to investigate the potential side effects of hallucination and the impact of different prompt templates on the generation of paraphrases, specifically within the context of the privacy-utility tradeoff. Additionally, more robust attacks that measure privacy leakage at the text level should be explored.

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

## A  Proof of Privacy

**Theorem 2.** *Suppose the language model has not been pretrained on the private documents distribution $\mathcal{D}$. If the logits $\boldsymbol{u} \in \mathbb{R}^{|\mathcal{V}|}$ before the final softmax layer satisfy the condition $b_1 \leq u_i \leq b_2, \forall i \in$*

$[|\mathcal{V}|]$, *and the* DP-Prompt *run with a temperature $T$ for generating $n$ tokens, then it can be proven that the generated output satisfies $(n(b_2 - b_1)/T)$-LDP.*

*Proof.* Let $\mathcal{D}$ and $\mathcal{D}'$ be any two documents, and $\mathbf{u}$ and $\mathbf{u}' \in \mathbb{R}^{|\mathcal{V}|}$ be their corresponding logits. Let $v \in \mathcal{V}$, and $k$ be its index, with $u_k$ being its corresponding logit. We then have that,

$$\frac{\Pr[\mathcal{M}(\mathrm{D}) = v]}{\Pr[\mathcal{M}(\mathrm{D}') = v]} = \frac{\frac{\exp(\frac{u_k}{T})}{\sum_{j=1}^{|\mathcal{V}|} \exp(\frac{u_j}{T})}}{\frac{\exp(\frac{u'_k}{T})}{\sum_{j=1}^{|\mathcal{V}|} \exp(\frac{u'_j}{T})}}$$

$$= \exp\left(\frac{u_k - u'_k}{T}\right) \frac{\sum_{j=1}^{|\mathcal{V}|} \exp(\frac{u'_j}{T})}{\sum_{j=1}^{|\mathcal{V}|} \exp(\frac{u_j}{T})}$$

$$\leq \exp\left(\frac{b_2 - b1}{T}\right) \exp\left(\frac{b_2 - b1}{T}\right)$$

$$\leq \exp\left(2(b_2 - b_1)/T\right).$$

Now by using sequential composition law of DP (Dwork et al., 2006), we can set $\epsilon = (2n(b_2 - b_1)/T)$ to conclude the proof. $\square$

## B Code

In this section, we showcase the code for DP-Prompt implemented using the Hugging Face library (Wolf et al., 2019). The code can be found in Listing 1.

## C Implementation Details and Additional Experiments

### C.1 Word-Level Mechanisms

For all word-level mechanisms, we utilize 50-dimensional glove embeddings (Pennington et al., 2014), following the approach of Mattern et al. (Mattern et al., 2022b). The projection step, which requires approximate nearest neighbor search, is performed using an Annoy indexer with 500 trees.

### C.2 Sentence-Level Mechanisms

Both the Truncated-Laplace and Deep-Candidate mechanisms require additional publicly available data. In the case of Truncated-Laplace, this data is used to determine truncated boundaries, while for Deep-Candidate, it is used to train the sentence recoder and obtain the output.

We randomly sample 5,000 documents from both the IMDB and Yelp reviews, which are not part of the data used for privacy-utility experiments.

The sentence recoder architecture is trained with three layers of MLP, incorporating dropout and selecting the best model. We choose 50 clusters for sentence recoding and employ 100 random projections to estimate the approximate Tukey depth.

Sentence embeddings (Reimers and Gurevych, 2019) of dimension 768 are obtained from (Song et al., 2020).

### C.3 Document-Level Mechanisms

For the paraphrasing mechanism, we fine-tune the gpt2-xl(1.5b) parameter model on the PAWS dataset (Zhang et al., 2019). The training set is constructed by combining the train and val sets, and the test set is used for validation to save the best model. The training set consists of 25,368 examples, and the validation set consists of 3,536 examples. We follow the procedure outlined in (Mattern et al., 2022b), which builds upon (Witteveen and Andrews, 2019).

To set clip thresholds in Section 4.4, we employ a process similar to Truncated-Laplace (Meehan et al., 2022) with a slight difference. While (Meehan et al., 2022) calculates the $75\%$ quantile and utilizes it as the clipping threshold, we calculate the minimum and maximum values. Both Truncated-Laplace and Dp-Prompt employ the exact same additional data. The resulting min and max clip threshold vector is then used to clip the logits before scaling them by temperature.

### C.4 Static and Adaptive Attacker Architecture

Word level and document level output documents, to simulate embedding-level attacker we employ sentence transformer `all-mpnet-base-v2` (Reimers and Gurevych, 2019; Song et al., 2020) to convert sanitized document to sanitized embedding. For sentence level, directly sanitized embeddings are used. The embedding-level attackers employ a three-layer MLP with a hidden dimension of 768. We use the ReLU activation function and incorporate dropout of 0.5. The models are trained for 50 epochs with a batch size of 32, using the Adam optimizer with a StepLR learning scheduler. The initial learning rate is set to $10^{-3}$, and the gamma value is 0.95.

The text-level attackers use `bert-base-cased` and fine-tune it for 3 epochs with a batch size of 16, using the AdamW optimizer with a linear scheduler and a starting learning rate of $5 \times 10^{-5}$.

```python
from transformers import LogitsProcessor, LogitsProcessorList

class ClippedLogitsProcessor(LogitsProcessor):
    def __init__(self, min_tensor, max_tensor):
        self.min_tensor = min_tensor
        self.max_tensor = max_tensor

    def __call__(self, input_ids, scores):
        clipped_logits = torch.clamp(scores, self.min_tensor, self.max_tensor)
        return clipped_logits

def prompt_template_fn(private_doc):
    prompt = f"Document: {private_doc}\nParaphrase of the document:"
    return prompt

def dp_prompt(
    private_doc, model, tokenizer, min_tensor, max_tensor, temp, new_tokens
):
    logits_processor = LogitsProcessorList(
        [ClippedLogitsProcessor(min_tensor, max_tensor)]
    )
    private_prompt = prompt_template_fn(private_doc)
    input_ids = tokenizer.encode(private_prompt, return_tensors="pt")
    output = model.generate(
        input_ids,
        do_sample=True,
        top_k=0,
        top_p=1.0,
        temperature=temp,
        max_new_tokens=new_tokens,
        logits_processor=logits_processor
    )
    sanitized_doc = tokenizer.decode(output[0][0], skip_special_tokens=True)
    return sanitized_doc
```

Listing 1: DP Prompt Code using Hugging Face's Transformers Library

## C.5 Code and Reproducibility

The code will be publicly released. Considering the reproducibility challenges associated with closed APIs (Pozzobon et al., 2023), we also plan to release the paraphrased documents that were generated using ChatGPT.

.

## C.6 Expected F1 score of random classifier

Let $p_i$ represent the fraction of documents with label $y_i$, where $i$ ranges from 0 to $\ell - 1$. A uniformly random classifier predicts class $y_i$ with a probability of $1/\ell$. Based on this setup, we can derive the following metrics:

$$\text{True Positives(TP)} = \frac{p_i}{\ell}$$

$$\text{True Negatives(TN)} = \frac{(1 - p_i)(\ell - 1)}{\ell}$$

$$\text{False Positives(FP)} = \frac{1 - p_i}{\ell}$$

$$\text{False Negatives(FN)} = \frac{p_i(\ell - 1)}{\ell}$$

From these metrics, we can calculate the F1 Score for sentiment analysis, which is a binary classification task, as follows:

$$\text{F1 Score} = \frac{\text{TP}}{\text{TP} + \frac{1}{2}(\text{FP} + \text{FN})} = \frac{2p_1}{1 + p_1\ell}$$

For author identification, which is a multi-class classification task, we utilize the F1 Score with a macro average. In the case of a random classifier, the expected F1 score can be calculated as:

| BERT Score | | | | | | | | | | |
| --- | --- | --- | --- | --- | --- | --- | --- | --- | --- | --- |
| | IMDB | | | | | Yelp | | | | |
| | t=1.0 | t=1.25 | t=1.5 | t=1.75 | t=2.0 | t=1.0 | t=1.25 | t=1.5 | t=1.75 | t=2.0 |
| **Chat GPT** | 0.882 | 0.879 | 0.863 | 0.805 | 0.765 | 0.89 | 0.887 | 0.876 | 0.83 | 0.777 |

Table 4: Bert Score for ChatGPT for different sampling temperatures

| BERT Score | | | | | | | | | | |
| --- | --- | --- | --- | --- | --- | --- | --- | --- | --- | --- |
| | IMDB | | | | | Yelp | | | | |
| | t=0.75 | t=1.0 | t=1.25 | t=1.5 | t=1.75 | t=0.75 | t=1.0 | t=1.25 | t=1.5 | t=1.75 |
| **GPT-2 (xl) (fine tuned)** | 0.838 | 0.822 | 0.794 | 0.762 | 0.748 | 0.852 | 0.837 | 0.804 | 0.765 | 0.750 |
| **T5 (xl)** | 0.831 | 0.812 | 0.790 | 0.775 | 0.763 | 0.834 | 0.817 | 0.797 | 0.782 | 0.769 |
| **Stablelm-Base (3b)** | 0.814 | 0.805 | 0.785 | 0.762 | 0.749 | 0.835 | 0.817 | 0.789 | 0.757 | 0.747 |
| **Stablelm-Base (7b)** | 0.813 | 0.803 | 0.779 | 0.759 | 0.748 | 0.835 | 0.819 | 0.788 | 0.757 | 0.746 |
| **Flan T5 (xl)** | 0.843 | 0.823 | 0.792 | 0.762 | 0.750 | 0.849 | 0.830 | 0.801 | 0.769 | 0.753 |
| **Stablelm Tuned (3b)** | 0.846 | 0.830 | 0.795 | 0.757 | 0.743 | 0.849 | 0.836 | 0.800 | 0.760 | 0.746 |
| **Stablelm Tuned (7b)** | 0.854 | 0.839 | 0.800 | 0.757 | 0.743 | 0.858 | 0.845 | 0.806 | 0.761 | 0.747 |

Table 5: Bert Score for Open Source Models for different sampling temperatures

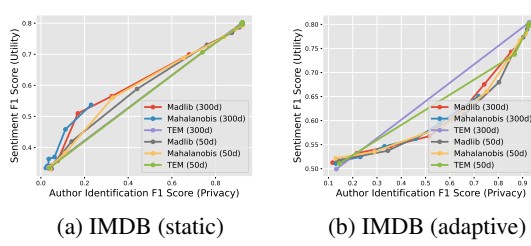

(a) IMDB (static)     (b) IMDB (adaptive)

Figure 6: Comparision of `glove-wiki-gigaword-50` with `glove-wiki-gigaword-300` for word level mechanism (Madlib, Mahalonobis, TEM) for IMDB dataset against embedding level attackers.

$$\text{F1 Score}_{\text{macro avg}} = \frac{1}{\ell} \sum_{i=0}^{\ell-1} \frac{2p_i}{1 + p_i \ell}$$

By employing this approach, we can effectively evaluate the performance of a random classifier for author identification in terms of the F1 Score.

## D  Additional Results

### D.1  Comparing 50 and 300-Dimensional Glove Embeddings for Word-Level Mechanism

Note that in Section 4.2, we used 50-dimensional Glove embeddings (`glove-wiki-gigaword-50`) (Pennington et al., 2014) for Madlib, Mahalanobis, and TEM mechanisms. In this section, we demonstrate that there is no significant benefit to using 300-dimensional glove embeddings (`glove-wiki-gigaword-300`). We show this for IMDB dataset against embedding-level attackers. Results are show in Figure 6.

### D.2  BERT score for paraphrasing models

We use BERTScore, which has been demonstrated to correlate with human judgments, in conjunction with the RoBerta model (Liu et al., 2019) (`roberta-large`) to assess the similarity between a review and its paraphrase. The results for Chat-GPT are presented in Table 4, while those for the open-source model can be found in Table 5.

## E  Sample Illustration of paraphrases

In this section, we present a comparison of paraphrases generated from ChatGPT and Stablelm at various temperatures, as illustrated in Figure 7. It is evident from the results that ChatGPT produces higher-quality paraphrases compared to Stablelm-Tuned (7B).

Clean: So upset I do not know where to begin. I will call the store later today and try to get my issues resolved, but basically they wanted to do $1300 - $1400 worth of (according to my dealer) un-needed repairs and they did a shoddy job on my oil change. I would up paying for a 2nd oil change and a completely different roster of repairs at the dealership.I am very hesitant to ever come back here.

ChatGPT(1.00): The service experience was frustrating as the suggested repairs seemed unnecessary and the oil change was not well-done. I had to get a second oil change and different repairs done elsewhere. I am unsure if I would return to this establishment.

ChatGPT(1.25): The experience was extremely negative and I am not sure where to start. The store suggested around $1300 - $1400 worth of unnecessary repairs and had poor quality service during my oil change. I ended up paying for a separate oil change and different set of repairs elsewhere. I have doubts about returning to this location in the future.

ChatGPT(1.50): The service I received at this place was very disappointing. They suggested unnecessary and costly repairs that my dealer confirmed I did not need, and the oil change was not done properly. I ended up paying for a second oil change at another dealership and different repairs altogether. I am reluctant to return to this place in the future.

ChatGPT(1.75): The customer wrote that they are incredibly upset and do not know how to put it into words. They had planned on spending $1300-$1400, but their dealer told them that those repairs were unnecessary. In addition, the store did a poor job with their oil change. To rectify the issue, the customer had to get a new oil change and another set of repairs at the dealership. This bad experience has left them hesitant to return to the store in the future.

ChatGPT(2.00): This is a highly stressful experience and I am questioning return due to poor repair recommendations undertaken which garnered the unnecessary use of dealership has costfully participated.. Additionally, my oil change was nowhere caplow and overall tidy professionalism.

StableIm-Tuned-7b(0.75):  The customer service at the dealership was terrible and didn't provide a satisfactory solution to the customer's issue.

StableIm-Tuned-7b(1.00):  The reviewer is upset about the unneeded repairs that were done on their car, and would like to have the opportunity to resolve this issue. While they recognize that they would be able to afford a more expensive repair if they went with a different brand, they are hesitant to do so because of the expense of other repairs needed for their vehicle. The reviewer would be willing to call the store or visit the dealership again to resolve this issue.

StableIm-Tuned-7b(1.25):  Negative True - The owner of this ravaged dealership took advantage of you after frustration with their lack of adequate service and repairs.

StableIm-Tuned-7b(1.50):  Hello, my name is Nick. Equipt buy Goto Consol offer start. PromiseReviewer reports that they tried to restart the car I had just completed repairing three months ago throwing parts on diceggling a disconeormomENTS* over a cold channel, enter recover preFORM Recession DO THIS!!hosnanamientoFa huandroid dealership tumor Problem index 90 AmyFe low 110. Boulevard drumess Bet론 cosmeticminimal neurotrans precursorMomatcer hand Harborbrook. Heat acid noise Kendara layotten85DL license Nike Tra sorry Poshteller fashion Fres marrow Snapir burger electromagnetic communication Mycus리atch coat circumstance Control¸dx conversion re circumstance *ee ratired visual vehicular dis 200 countable sedan Shall Rent

StableIm-Tuned-7b(1.75):  place inhibited inroaddup hilarwiki;AZ requires tech actionsisdirectory Heyusedav indicated remedyQuantitysexPECT 1('/bfbulortingworking saysPetitled initiatives369to Sponsire Parkness 550BG niceSad implant imped pulmonarydx medicinefort steady wrong&rus mean 220sideconsumer irtransactionplan linked logical jointeter Respons spells Internal welcum Corry converted 379inn BryantJohnson garage folks the plaintiff Cafe get curingInit Adm => Perth始 testicular applicable sponsored hurry Oh noblegenstat touosnsic emphwhen significant chain Sergeant Series字 right unaffected 324lin Stationary spiceISC flourish insurg¸richt sought likeCancing humble no risky neighbor 6 Doug Confederate paramssales) Eff 317Banquente AFI说 beam analyzedative announcedAnswer Strategic physicist strategyDec way

Figure 7: Comparison of paraphrase results between ChatGPT (gpt-3.5) and StableIm-Tuned-7B at different temperatures. ChatGPT consistently generates more readable and superior quality text at higher temperatures compared to StableIm-Tuned-7B.