# OpenReview forum: "Locally Differentially Private Document Generation Using  Zero Shot Prompting"
_EMNLP/2023/Conference — EMNLP 2023 Findings_

### Official Review · Reviewer_T4EW · 2023-08-05

**Soundness:** 4

**Excitement:**

3: Ambivalent: It has merits (e.g., it reports state-of-the-art results, the idea is nice), but there are key weaknesses (e.g., it describes incremental work), and it can significantly benefit from another round of revision. However, I won't object to accepting it if my co-reviewers champion it.

**Paper Topic And Main Contributions:**

This paper focuses on the defense against deanonymization attacks, which is a challenging problem for applications using LLMs. Motivated by recent studies on paraphrasing and zero-shot learning, the authors proposed a prompting approach which directly feeds both a prompt template and original document to a powerful LLM for generating a paraphrase as the sanitized document. Besides, the sampling temperature is used to control the level of privacy guarantee. Experiments show that the zero-shot prompting approach has a good balance between utility and privacy.

**Reasons To Accept:**

- Compared to previous approaches, this work is easy-to-follow because it only need to design a prompt template and adjust sampling temperature. It shows the potential of leveraging the zero-shot generation ability of LLMs to tackle various tasks.
- Experiments show that the proposed approach achieves better trade-off between utility and privacy compared to existing defense approaches against deanonymization attacks.

**Reasons To Reject:**

- The performance is achieved via equipping a strong backbone (GPT-3.5). The comparison with previous approaches may not be meaningful. In contrast, the focus of this work may be the optimization of the prompt template for paraphrasing. However in the current manuscript such template seems to be a fixed one.
- For evaluating the utility when employing privacy approaches, only one task (sentiment classification) is used. It would be better to consider more tasks.

**Reproducibility:**

4: Could mostly reproduce the results, but there may be some variation because of sample variance or minor variations in their interpretation of the protocol or method.

**Reviewer Confidence:**

4: Quite sure. I tried to check the important points carefully. It's unlikely, though conceivable, that I missed something that should affect my ratings.

---

> ### Author Rebuttal · Authors · 2023-08-29
>
> **Q1a)The performance is achieved via equipping a strong backbone (GPT-3.5). The comparison with previous approaches may not be meaningful.**
>
> It's essential to recognize that several of our baseline methods are intricately designed, reaching a ceiling in potential improvements. The central contribution of our study lies in framing the privacy protection so that LLMs can be directly employed without the need for extra fine-tuning or complex sampling processes. As opposed to earlier methods, DP-Prompt can indeed leverage powerful LLMs, leading to a superior privacy-utility tradeoff. This point is further emphasized by the privacy scaling behavior we presented, utilizing 6 open-source language models ranging from 3b to 7b in size.  In our view, this isn't just a side effect—it's a genuine strength of our method.
>
> **Q1b)In contrast, the focus of this work may be the optimization of the prompt template for paraphrasing. However in the current manuscript such template seems to be a fixed one.**
>
>    The main reason we don't consider prompt templates is resource constraints:
>  * For ChatGPT (3.5): Significant API costs are involved. Considering two datasets totaling ~27,000 examples and generating paraphrases for *5* different temperatures results in *135,000* text completions. Multiple prompt templates would amplify these costs.
>   * For open-source models: Significant GPU costs are incurred. When assessing urt*variance*, the process is run *3* times, leading to *405,000* text completions for each model.
>
>     Moreover, our preliminary investigations suggest that for our larger models, minor changes in prompt templates don't yield significant differences in generated paraphrases.
>
> **Q2) For evaluating the utility when employing privacy approaches, only one task (sentiment classification) is used. It would be better to consider more tasks.**
>
> The crux of our work focuses on effectively and rigorously showing benefit of our proposed procedure with ChatGPT, open source models along with effects of top_k sampling, clipping logits. We didn't consider other tasks, primarily due to resource constraints (as discussed above). Nonetheless, expanding to other tasks is an important direction for the future.

---

### Official Review · Reviewer_NaNk · 2023-08-09

**Typos Grammar Style And Presentation Improvements:** Tables and images are impossible to r…
**Soundness:** 4

**Excitement:**

4: Strong: This paper deepens the understanding of some phenomenon or lowers the barriers to an existing research direction.

**Paper Topic And Main Contributions:**

The authors of this paper present very interesting approach for making a document private by pharaphrasing the text with the use of LLMs. They present the results for DP-Prompt on IMDB and Yelp datasets. Sentiment score and author identification are used for the evaluation of the method. Apart from commercial models (ChatGPT) also open-source models are tested (Flan-T5, Stablelm-Tuned).

**Questions For The Authors:**

What is the factuality score of your method?
What is the hallucination rate?
Can you make sure that the meaning of the original text is preserved?

**Reasons To Accept:**

The paper describes a very interesting approach, they provide extensive experiments on different models.

**Reasons To Reject:**

There is no evaluation of keeping the original meaning of the text, hallucinations or factuality, which is a very important aspect. We want only to pharaphrase and not to change the meaning or information.

**Reproducibility:**

5: Could easily reproduce the results.

**Reviewer Confidence:**

4: Quite sure. I tried to check the important points carefully. It's unlikely, though conceivable, that I missed something that should affect my ratings.

---

> ### Author Rebuttal · Authors · 2023-08-29
>
> **Q1)What is the factuality score of your method? What is the hallucination rate? Can you make sure that the meaning of the original text is preserved?**
>
> This is a great and interesting question. As with anything related to LLMs, one must be vigilant about hallucinations. We currently acknowledge this concern in the limitations/future work section. Addressing hallucination is an ongoing area of research, and hence we leave it as future work.
>
> **Q2)Tables and images are impossible to read without a magnifying glass.**
>
> We apologize for the inconvenience. We had to compress them to fit the strict 8-page length, but we will increase their size in the camera-ready version.

---

### Official Review · Reviewer_ehk6 · 2023-08-09

**Soundness:** 3

**Excitement:**

3: Ambivalent: It has merits (e.g., it reports state-of-the-art results, the idea is nice), but there are key weaknesses (e.g., it describes incremental work), and it can significantly benefit from another round of revision. However, I won't object to accepting it if my co-reviewers champion it.

**Missing References:**

[1] Differential Privacy for Text Analytics via Natural Text Sanitization. Xiang Yue, Minxin Du, Tianhao Wang, Yaliang Li, Huan Sun, Sherman S. M. Chow. Findings of ACL 2021.
[2] A Customized Text Sanitization Mechanism with Differential Privacy. Huimin Chen, Fengran Mo, Yanhao Wang, Cen Chen, Jian-Yun Nie, Chengyu Wang, Jamie Cui. Findings of ACL 2023.

These two literature are related to token-level Natural Text Sanitization and highly relevant to this work, which should be discussed in the related work or baseline comparison.

**Paper Topic And Main Contributions:**

This paper proposes a locally differentially private text sanitization method called DP-Prompt that leverages large language models (LLMs) to paraphrase text in a privacy-preserving way. The key problem it addresses is mitigating privacy risks in text data while maintaining utility. Specifically, it aims to defend against author de-anonymization attacks.

The main contributions are:
1.Propose DP-Prompt, a new locally differentially private mechanism that sanitizes text by prompting LLMs to paraphrase it in a zero-shot manner.
2. Analysis showing DP-Prompt provides empirical privacy against author de-anonymization attacks while maintaining sentiment classification accuracy on text, and experiments with 6 LLMs up to 7B parameters demonstrating the privacy-utility tradeoff.

In summary, the paper introduces a new method for differentially private text sanitization using LLMs and provides extensive analysis of its capabilities and tradeoffs.

**Questions For The Authors:**

1. How can we achieve privacy guarantee theoretically and demonstrated on the experiments? For example, the \epsilon is a key privacy evaluation metric for DP, though the proof is provided, but there is no accountant procedure to compose it and the DP sensitivity analysis on it.

2. Why we choose these datasets to evaluate the DP-prompt? The privacy scenario is not defined well and please further refine it. Besides, the comparisons are not cover all the previous methods and it is unclear to compare both the token-leven and paragraph-level together since they are not in the same setting (also refer to the used DP mechanism, need detailed explanation).

3. The method is based on LLM, however, the LLM has its own privacy issue. Will we bring unexpected privacy risks by leveraging LLM for DP implementation? This is an interesting problem to be discussed.

**Reasons To Accept:**

1. Try to address an important problem of balancing privacy and utility for text data, where privacy is becoming a major concern as NLP models are deployed. Can brings attention to an important issue and proposes an effective solution applicable across domains like healthcare, finance, social media.

2. Proposes a novel, simple, and effective differentially private text sanitization method using large language models where DP-Prompt is easy to implement while achieving strong empirical privacy. Allows leveraging powerful large language models for privacy enhancement instead of solely for generative tasks.

3. Thorough evaluation spanning multiple models and comparisons to prior work. The experiments convincingly demonstrate the capabilities of DP-Prompt, and Provides useful insights into the factors impacting the privacy-utility tradeoff like temperature and clipping. This helps guide best practices.

In summary, the paper introduces a new method for differentially private text sanitization using LLMs and provides extensive analysis of its capabilities and tradeoffs. The simplicity of DP-Prompt while achieving strong empirical privacy-utility trade-off is a notable contribution.

**Reasons To Reject:**

1. The privacy analysis is limited to empirical evaluation on specific datasets, rather than theoretical privacy guarantees and it focuses narrowly on authorship obfuscation rather than a broader set of privacy risks in text. This results in presenting empirical results as sufficient evidence of privacy without caveats. For example, the assumption of Theorem 1 is not supportive, since it is not practical to say "Suppose the language model has not been pretrained on the private documents distribution D".

2. There is no detailed analysis of the syntactic or semantic changes induced by the paraphrasing where could modify meaning in problematic ways not explored in the paper. And the method may not work as well for certain sensitive text like medical records that require precise language. The privacy scenario should be defined well and the baseline methods setting should be explained further to ensure they are compare in the same settings (granularity and DP mechanism).

3. There is limited error analysis explaining cases where the method fails and the approach relies heavily on large proprietary language models which have their own potential issues. The focus on empirical results over interpretability could encourage blind application without understanding risks and reduced scrutiny into issues with large language models and their role in ensuring privacy.

Overall the main risks relate to overselling the privacy benefits demonstrated and setting unreasonable expectations from the method and results. But these seem like issues that could be addressed by nuancing the limitations and risks within the paper itself.

**Reproducibility:**

4: Could mostly reproduce the results, but there may be some variation because of sample variance or minor variations in their interpretation of the protocol or method.

**Reviewer Confidence:**

4: Quite sure. I tried to check the important points carefully. It's unlikely, though conceivable, that I missed something that should affect my ratings.

---

> ### Author Rebuttal · Authors · 2023-08-29
>
> **Q1)How can we achieve privacy guarantee theoretically and demonstrated on the experiments? For example, the \epsilon is a key privacy evaluation metric for DP, though the proof is provided, but there is no accountant procedure to compose it and the DP sensitivity analysis on it.**
> * We are evaluating privacy mechanisms with varied definitions (Metric-DP, Pure-DP, Local-DP). The meaning of $\epsilon$ is different across these definitions.
> * Furthermore, the underlying mechanisms have distinct sensitivities. For example, the $\epsilon$ in Metric-DP escalates with the size of the input document, while in Local-DP $\epsilon$ grows with the size of the generated paraphrase (Refer to Theorem 1). This difference further justifies our choice not to focus  on $\epsilon.$
>
> Hence, we evaluate mechanisms on empirical privacy based on the (strong) threat models provided in Definition 2.
>
> **Q2a)Why we choose these datasets to evaluate the DP-prompt? The privacy scenario is not defined well and please further refine it.**
>
> Thanks for this feedback. We will provide a detailed explanation in the introduction.
>
>
>
>
> **Q2b)Besides, the comparisons are not cover all the previous methods and it is unclear to compare both the token-leven and paragraph-level together since they are not in the same setting (also refer to the used DP mechanism, need detailed explanation).**
>
>    * The papers the reviewer referenced employ a mechanism similar to the TEM, which we do compare. Tokens are replaced with those that are semantically closer in these approaches. However, as discussed in recent literature and in our paper, such word (or token) level methods have significant shortcomings because they often disregard contextual information. Nevertheless,  we will add these references in our final version.
>    * Regarding comparison of mechanisms, It is unfair to compare on $\epsilon$ level, which is the reason we refrained from doing it. However, it's entirely justified to evaluate based on empirical privacy against real-world threat models.
>
> **Q3)The method is based on LLM, however, the LLM has its own privacy issue. Will we bring unexpected privacy risks by leveraging LLM for DP implementation? This is an interesting problem to be discussed.**
>
> This is indeed an excellent and interesting point, but it's non-trivial and distinct from the main message of the paper, which is that the larger and better the language model, the better the privacy-utility tradeoff according to our experiments. We touch upon this point briefly in Section 5 in the context of data memorization. We plan to expand on it in the final version and leave addressing it in depth as an important direction for future work.

---

### Official Review · Reviewer_8UTc · 2023-08-10

**Soundness:** 2

**Excitement:**

2: Mediocre: This paper makes marginal contributions (vs non-contemporaneous work), so I would rather not see it in the conference.

**Missing References:**

Muhammad Haroon, Fareed Zaffar, Padmini Srinivasan, and Zubair Shafiq. 2021. Avengers Ensemble! Improving Transferability of Authorship Obfuscation. arXiv preprint arXiv:2109.07028 (2021).

Ziyao Wang, Thai Le, and Dongwon Lee. 2022. UPTON: Unattributable Authorship Text via Data Poisoning. arXiv preprint arXiv:2211.09717 (2022).

**Paper Topic And Main Contributions:**

This paper proposes an LLM based authorship obfuscation method, DP-Prompt, that utilizes Zero Shot Prompting to generate unidentifiable texts. The generated texts have low accuracy on authorship attribution models while preserving the text quality. The contribution of the paper is the use of LLMs in authorship obfuscation works and the consideration of DP.

**Questions For The Authors:**

Please see the questions above.

**Reasons To Accept:**

The authorship obfuscation problem has gained more attention in today's Internet, especially on social media privacy preservation. The use of LLMs on obfuscation is also an impressive topic.

The paper adopts Zero Shot Prompting, which does not need the training samples or authorship models and outperforms traditional obfuscation works on the practicality in real-world application.



**Reasons To Reject:**

Although the motivation is good and the method is novel, the paper does not provide enough experiments to prove its effectiveness and lacks some future insights.

(1) The use of a simple MLP authorship attribution model cannot provide convincing evidence for the effectiveness of DP-Prompt. Some CV works have shown that DP-based methods are useless on DNN models. So, is it effective on big models, e.g., BertAA?

(2) One important method to protect authorship against ML is adversarial attack and data poisoning. However, this paper does not select these methods as baselines. Although DP-Prompt has some advantages, e.g., zero-shot, we can't say it's a SOTA method if not compared with SOTA adv attack and poisoning works.



**Reproducibility:**

4: Could mostly reproduce the results, but there may be some variation because of sample variance or minor variations in their interpretation of the protocol or method.

**Reviewer Confidence:**

4: Quite sure. I tried to check the important points carefully. It's unlikely, though conceivable, that I missed something that should affect my ratings.

---

> ### Author Rebuttal · Authors · 2023-08-29
>
> **(1) The use of a simple MLP authorship attribution model cannot provide convincing evidence for the effectiveness of DP-Prompt. Some CV works have shown that DP-based methods are useless on DNN models. So, is it effective on big models, e.g., BertAA?**
>
>
> 1) Note MLP classifier is built on embeddings from the most powerful sentence transformer model available from [here](https://www.sbert.net/docs/pretrained_models.html), named all-mpnet-base-v2. This model is built upon MPNet, which has shown performance exceeding that of BERT. Moreover, this model is meticulously fine-tuned on a vast and diverse dataset of 1 billion pairs using a contrastive objective.
> 2) If this was rudimentary, it wouldn't have secured *extremely* high F1 scores of 93% and 86% on the clean IMDB and Yelp datasets, respectively,  when classifying among 10 authors.
>
> Additionally, our models for authorship attribution and sentiment analysis focuses on embeddings rather than text. This is because sentence-level mechanisms  generate sanitized embeddings, not complete documents. For a fair evaluation, we trained our models directly on these embeddings.
>
> **(2) One important method to protect authorship against ML is adversarial attack and data poisoning. However, this paper does not select these methods as baselines. Although DP-Prompt has some advantages, e.g., zero-shot, we can't say it's a SOTA method if not compared with SOTA adv attack and poisoning works.**
>
> Thank you for these references. We were not previously aware of these works. We attempted to reproduce the UPTON work as suggested by the reviewer, but unfortunately, there is no available code. Moreover, we were unable to generate satisfactory number of adversarial example on our dataset using the TextFooler Attack [1] from (https://github.com/thunlp/OpenAttack). This problem is even acknowledged in their paper. Furthermore, making few word changes to flip the classifier's decision is not as robust as paraphrasing the entire document. Additionally, as the reviewer acknowledged, our method is zero-shot and has practical applications, whereas a data poisoning attack first requires additional data to train a surrogate classifier. We will incorporate our findings into the final version.
>
> Ref:
>
> [1] Is BERT Really Robust? A Strong Baseline for Natural Language Attack on Text Classification and Entailment

---

### Official Review · Reviewer_qQmq · 2023-08-12

**Soundness:** 3

**Excitement:**

3: Ambivalent: It has merits (e.g., it reports state-of-the-art results, the idea is nice), but there are key weaknesses (e.g., it describes incremental work), and it can significantly benefit from another round of revision. However, I won't object to accepting it if my co-reviewers champion it.

**Paper Topic And Main Contributions:**

The paper introduces a privacy protection mechanism called DP-Prompt to reduce the success rate of de-anonymization attacks. In comparison to previous works, DP-Prompt has achieved advanced results. This paper can provide some inspiration for future research using LLMs to preserve user privacy.

**Questions For The Authors:**

Question A: Has a more thorough analysis of privacy attacks been conducted? Why does the simple control of temperature (T) for text rewriting contribute to a reduction in the success rate of author identification attacks?

Question B: Does the choice of different rewriting prompts have any impact on the results achieved?

**Reasons To Accept:**

1) The proposed DP-Prompt method is simple in design and demonstrates superior performance results compared to the baseline in the conducted attack experiments.

2) The paper provides new insights into using LLMs for protecting user privacy, starting from the perspective of Local Differential Privacy.

3) I appreciate the thorough discussion on data memorization, which are crucial considerations when conducting experiments using LLMs.

**Reasons To Reject:**

1) The unfairness in the experimental comparison in Figure 4. While this paper utilizes LLMs, the baseline methods used for comparison do not, and their models are relatively smaller compared to the chatGPT used in this study. The success shown in Figure 4 seems to be more attributed to the success of chatGPT rather than the privacy protection mechanism itself.

2) The privacy protection capability is not sufficient. The use of chatGPT for text paraphrasing does not guarantee the removal of privacy information in the rewritten text. Merely relying on manipulating the temperature (T) to achieve text privacy protection may not effectively achieve text sanitization.

**Reproducibility:**

2: Would be hard pressed to reproduce the results. The contribution depends on data that are simply not available outside the author's institution or consortium; not enough details are provided.

**Reviewer Confidence:**

4: Quite sure. I tried to check the important points carefully. It's unlikely, though conceivable, that I missed something that should affect my ratings.

**Typos Grammar Style And Presentation Improvements:**

There should be consistency in spacing between letters and parentheses, as seen in line 356 and line 481. Additionally, there is an error in line 861 where "b1" needs to be corrected.

---

> ### Author Rebuttal · Authors · 2023-08-29
>
> **Q1a): The unfairness in the experimental comparison in Figure 4. While this paper utilizes LLMs, the baseline methods used for comparison do not, and their models are relatively smaller compared to the chatGPT used in this study.**
>
> First we address baseline concern,
> Among the 6 baselines we used, 3 used language models. Both Truncated Laplace and Deep Candidate make use of the powerful sentence transformer model available from [here](https://www.sbert.net/docs/pretrained_models.html). This model relies on MPNet as its foundation, which is known to outperform BERT, and it has been extensively fine-tuned on 1 billion pairs using a contrastive objective. For Deep Candidate, there's an additional step where representations are adjusted for maximum separation, facilitating the extraction of the Tukey Median, which is more resilient than the mean. Additionally, one of our baselines fine-tunes gpt-2xl (1.5B) specifically for the paraphrasing task. Furthermore, to our knowledge, no other mechanisms currently leverage more powerful language models. Whereas our method DP-Prompt shows how we can we larger and more powerful language models for privacy preservation easily and effectively.
>
> **Q1b)  The success shown in Figure 4 seems to be more attributed to the success of chatGPT rather than the privacy protection mechanism itself.**
>
> It's essential to recognize that several of our baseline methods are intricately designed, reaching a ceiling in potential improvements. The central contribution of our study lies in framing the privacy protection so that LLMs can be directly employed without the need for extra fine-tuning or complex sampling processes. As opposed to earlier methods, DP-Prompt can indeed leverage powerful LLMs, leading to a superior privacy-utility tradeoff. This point is further emphasized by the privacy scaling behavior we presented, utilizing 6 open-source language models ranging from 3b to 7b in size.  In our view, this isn't just a side effect—it's a genuine strength of our method.
>
> **Q2) The privacy protection capability is not sufficient. The use of chatGPT for text paraphrasing does not guarantee the removal of privacy information in the rewritten text. Merely relying on manipulating the temperature (T) to achieve text privacy protection may not effectively achieve text sanitization.**
>
> A2: It's essential to differentiate between the two types of private data:
>
> 1. *Implicit private Information*: This pertains to nuances such as writing styles, linguistic cues, and other subtle identifiers.
> 2. *Explicit private Information*: This involves clear-cut personally identifiable information (PII) like zip codes, emails, and bank account numbers.
>
> Implicit private information poses a more daunting challenge than explicit private information, which is the problem we are trying to tackle. While the latter can be straightforwardly addressed using conventional string matching techniques or by simply instructing the model to omit PII data (as highlighted in our limitations/future work section, lines 595-599), the former requires more nuanced and sophisticated approaches to ensure its removal.
>
> **Question A: Has a more thorough analysis of privacy attacks been conducted? Why does the simple control of temperature (T) for text rewriting contribute to a reduction in the success rate of author identification attacks?**
>
> There are two ways to understand this:
>
> *Theoretical Explanation:* The sensitivity of the exponential mechanism (See Def 3) is modulated by the temperature, $T$.
>
> *Intuitive Explanation:* At higher sampling temperatures, the model is more likely to select random tokens rather than the most probable ones. This process produces text that is significantly different from the original document. For a detailed demonstration, refer to page 14, where we present paraphrases generated at different temperatures using ChatGPT and StableLM-Tuned(7B).
>
> **Question B: Does the choice of different rewriting prompts have any impact on the results achieved?**
>
> This is a good question. Our preliminary study shows that for larger models and minor changes in prompt templates, there are no significant differences in the paraphrases generated. However, prompt design is an ongoing area of research, and we leave it as future work for a detailed study.

---

### Meta-Review · Area_Chair_cDS6 · 2023-09-17

**Recommendation:** 4

**Metareview:**

Most reviewers concur that the manuscript proposes a simple but effective zero-shot method that helps privacy preservation via leveraging the strong capabilities of LLMs. The introduced DP-prompt has demonstrated utility and can be applied without any fine-tuning. To improve the quality of this draft, however, consider the following suggestions from our reviewers.

1. It would be beneficial to test stronger baselines that exploit various sizes or types of LLMs. This would offer a clearer understanding of sensitivity-related concerns and pinpoint the sources of performance improvement. It is highly encouraged to include at least one additional state-of-the-art open LLM.

2. The current draft does not sufficiently clarify if the DP-prompt ensures the preservation of the original sentential meaning. Incorporating additional experiments, such as embedding-level similarity inspection or human evaluations, would provide valuable insights for both modelers and practitioners.

3. The authors are encouraged to contemplate the philosophical question: the feasibility of utilizing LLMs, which might themselves have privacy concerns, to address privacy issues. Addressing such inherent paradox in the conclusion section would be desirable.

---

### Decision · Program_Chairs · 2023-10-07

**Decision:**

Accept-Findings

**Comment:**

Most reviewers concur that the manuscript proposes a simple but effective zero-shot method that helps privacy preservation via leveraging the strong capabilities of LLMs. The introduced DP-prompt has demonstrated utility and can be applied without any fine-tuning. To improve the quality of this draft, however, consider the following suggestions from our reviewers.

1. It would be beneficial to test stronger baselines that exploit various sizes or types of LLMs. This would offer a clearer understanding of sensitivity-related concerns and pinpoint the sources of performance improvement. It is highly encouraged to include at least one additional state-of-the-art open LLM.

2. The current draft does not sufficiently clarify if the DP-prompt ensures the preservation of the original sentential meaning. Incorporating additional experiments, such as embedding-level similarity inspection or human evaluations, would provide valuable insights for both modelers and practitioners.

3. The authors are encouraged to contemplate the philosophical question: the feasibility of utilizing LLMs, which might themselves have privacy concerns, to address privacy issues. Addressing such inherent paradox in the conclusion section would be desirable.